# Multiplexing rhythmic information by spike timing dependent plasticity

**Nimrod Sherf**[1,2]*, **Maoz Shamir**[1,2,3]

**1** Physics Department, Ben-Gurion University of the Negev, Beer-Sheva, Israel, **2** Zlotowski Center for Neuroscience, Ben-Gurion University of the Negev, Beer-Sheva, Israel, **3** Department of Physiology and Cell Biology Faculty of Health Sciences, Ben-Gurion University of the Negev, Beer-Sheva, Israel

* sherfnim@post.bgu.ac.il

## Abstract

Rhythmic activity has been associated with a wide range of cognitive processes including the encoding of sensory information, navigation, the transfer of information and others. Rhythmic activity in the brain has also been suggested to be used for multiplexing information. Multiplexing is the ability to transmit more than one signal via the same channel. Here we focus on frequency division multiplexing, in which different signals are transmitted in different frequency bands. Recent work showed that spike-timing-dependent plasticity (STDP) can facilitate the transfer of rhythmic activity downstream the information processing pathway. However, STDP has also been known to generate strong winner-take-all like competition between subgroups of correlated synaptic inputs. This competition between different rhythmicity channels, induced by STDP, may prevent the multiplexing of information. Thus, raising doubts whether STDP is consistent with the idea of multiplexing. This study explores whether STDP can facilitate the multiplexing of information across multiple frequency channels, and if so, under what conditions. We address this question in a modelling study, investigating the STDP dynamics of two populations synapsing downstream onto the same neuron in a feed-forward manner. Each population was assumed to exhibit rhythmic activity, albeit in a different frequency band. Our theory reveals that the winner-take-all like competitions between the two populations is limited, in the sense that different rhythmic populations will not necessarily fully suppress each other. Furthermore, we found that for a wide range of parameters, the network converged to a solution in which the downstream neuron responded to both rhythms. Yet, the synaptic weights themselves did not converge to a fixed point, rather remained dynamic. These findings imply that STDP can support the multiplexing of rhythmic information, and demonstrate how functionality (multiplexing of information) can be retained in the face of continuous remodeling of all the synaptic weights. The constraints on the types of STDP rules that can support multiplexing provide a natural test for our theory.

**Data Availability Statement:** All relevant data are within the manuscript and its Supporting Information files.

**Funding:** This work was supported in part by: The Israel Science Foundation (ISF) grant number 300/ 16 and the National Science Foundation under

grant no. NSF PHY-1748958 and the United States-Israel Binational Science Foundation grant 2013204. The funders had no role in study design, data collection and analysis, decision to publish, or preparation of the manuscript.

**Competing interests:** The authors have declared that no competing interests exist.

## Author summary

Spike timing dependent plasticity (STDP) quantifies the change in the synaptic efficacy as a function of the temporal relationship between pre- and post-synaptic firing. STDP can be viewed as a microscopic unsupervised learning rule, and a wide range of such microscopic learning rules have been described empirically. Since there is no supervisor in unsupervised learning (which would provide with the system its goal), theoreticians have struggled with the question of the possible computational roles of the various STDP rules. Previous studies have focused on the possible contribution of STDP to the spontaneous development of spatial structure. However, the rich temporal repertoire of reported STDP rules has largely been ignored. Here we studied the contribution of STDP to the development of temporal structure. We show how STDP can shape synaptic efficacies to facilitate the transfer of rhythmic information downstream and to enable the multiplexing of information across different frequency channels. Our work emphasizes the relationship between the temporal structure of the STDP rule and the rhythmic activity it can support.

## Introduction

Neuronal oscillations have been described and studied for more than a century [1–14]. Rhythmic activity in the central nervous system has been associated with: attention, learning, encoding of external stimuli, consolidation of memory and motor output [8–10, 12, 14–23]. Rhythmic activity has also been suggested to support multiplexing in the central nervous system [24, 25]. Multiplexing is the ability to transmit two or more different signals via the same channel. The two main forms of multiplexing are: (*i*) Time division multiplexing, in which different time slots are allocated for the transmission of the different signals, e.g., distributing information for different clients by the same server. (*ii*) Frequency division multiplexing, in which different signals are transmitted via different frequency bands, e.g., the allocation of different frequency bands for different radio stations.

Various forms of multiplexing have been proposed to be utilized by the brain [24–34]. Caruso et al. [32] suggested that time division multiplexing is used in the auditory system to represent different objects. They demonstrated that some neurons shift their response from one object to another in time (similar to fluctuating focus of attention) in a manner that correlated with behaviour. Frequency division multiplexing has been suggested by Teng & Peoppel [30] (they also suggested other methods as well) to be utilized by the auditory system for encoding different features of the same auditory object in the theta and gamma channels; thus, frequency division multiplexing may also be used for binding. Here, we focus on frequency division multiplexing. Our aim is to study a mechanism that can shape synaptic connectivity to facilitate frequency division multiplexing.

The transfer of even a single oscillatory signal downstream is not necessarily trivial and requires some mechanism to prevent destructive interference and maintain the rhythmic component [35]. Recently, it has been suggested that synaptic plasticity, and especially spike-timing-dependent plasticity (STDP), can provide such a mechanism. STDP can be thought of as a generalization of Hebb's rule that neurons that "fire together wire together" [36] to the temporal domain. In STDP, the amount of potentiation and depression depends on the temporal relation between the pre- and post-synaptic firing [2, 37–44]. Luz and Shamir [35] analyzed the characteristics of the STDP rule that will enable the transfer of a single frequency channel downstream.

Multiplexing requires the transfer of more than one frequency channel. However, STDP has been shown to generate a winner-take-all like competition between subgroups of correlated pools of neurons [45–49]. Consequently, one may expect that the transfer of one frequency channel will suppress the other; thus, raising serious doubts whether STDP is consistent with multiplexing in the brain. Can STDP develop the capacity for transmitting rhythmic activity in more than one frequency band spontaneously, and facilitate multiplexing of information?

We address this question here in the framework of a modelling study. Below, we define the network architecture and the STDP learning rule. We then derive a mean-field approximation for the STDP dynamics in the limit of slow learning rate for a threshold-linear Poisson downstream neuron model. Analysing the STDP dynamics yields constraints on the STDP rules that enable multiplexing. Next, we test the generalisation of our understanding beyond the simplified analytical toy model using numerical simulations. Finally, we summarize our results and discuss how STDP can yield robustness of function in the face of constant synaptic remodelling.

## Results

### The pre-synaptic populations

We model a system of two excitatory populations of $N$ neurons, each responding to a different feature of an external stimulus, Fig 1. The external stimulus is characterized by two feature variables, $D_1$ and $D_2$, to which populations 1 and 2 respond, respectively. The response of each population is further assumed to be rhythmic, albeit in a different frequency, representing the different features of the stimulus.

The spiking activity of neuron $k \in \{1, \ldots N\}$ in population $\eta \in \{1, 2\}$, $\rho_{\eta,k}(t) = \sum_i \delta(t - t_{k,i}^\eta)$ (where $\{t_{k,i}^\eta\}_{i=1}^\infty$ are the spike times) is a doubly stochastic process. Given the 'intensity' $D_\eta$ of feature $\eta \in \{1, 2\}$ of the external stimulus, the spiking activity, $\rho_{\eta,k}(t)$, follows an independent inhomogeneous Poisson process statistics with a mean rate (mean over the Poisson distribution given the intensity variables $D_1$ and $D_2$) that is given by:

$$\langle \rho_{\eta,k}(t) \rangle = D_\eta(1 + \gamma \cos[\nu_\eta t - \phi_{\eta,k}]), \quad \phi_{\eta,k} = 2\pi k/N. \tag{1}$$

where $\nu_\eta$ is the angular frequency of oscillations for neurons in population $\eta$, $\gamma$ is the modulation to the mean ratio of the firing rate, and $\phi_{\eta,k}$ is the preferred phase of the $k$th neuron from population $\eta$. The preferred phases are assumed to be evenly spaced on the ring. Thus, it is convenient to think of the neurons in each population as organized on a ring according to their preferred phases of firing.

As the intensity parameters, $D_\eta$, represent features of the external stimulus they fluctuate on a timescale which is typically longer than the characteristic timescale of the neural response. For simplicity we assumed that $D_1$ and $D_2$ are independent random variables with identical distributions:

$$\langle D_\eta \rangle = D,$$
$$\langle D_\eta D_\xi \rangle = D^2(1 + \sigma^2 \delta_{\eta\xi}), \tag{2}$$

where $\langle \ldots \rangle$ denotes averaging with respect to the neuronal noise and stimulus statistics. The essence of multiplexing is to enable the transmission of different information channels; hence, the assumption of independence of $D_1$ and $D_2$ represents fluctuations of different features. This assumption also drives the winner-take-all competition between the two populations. We further assume that the stimulus changes at a slower timescale than the neural responses; thus,

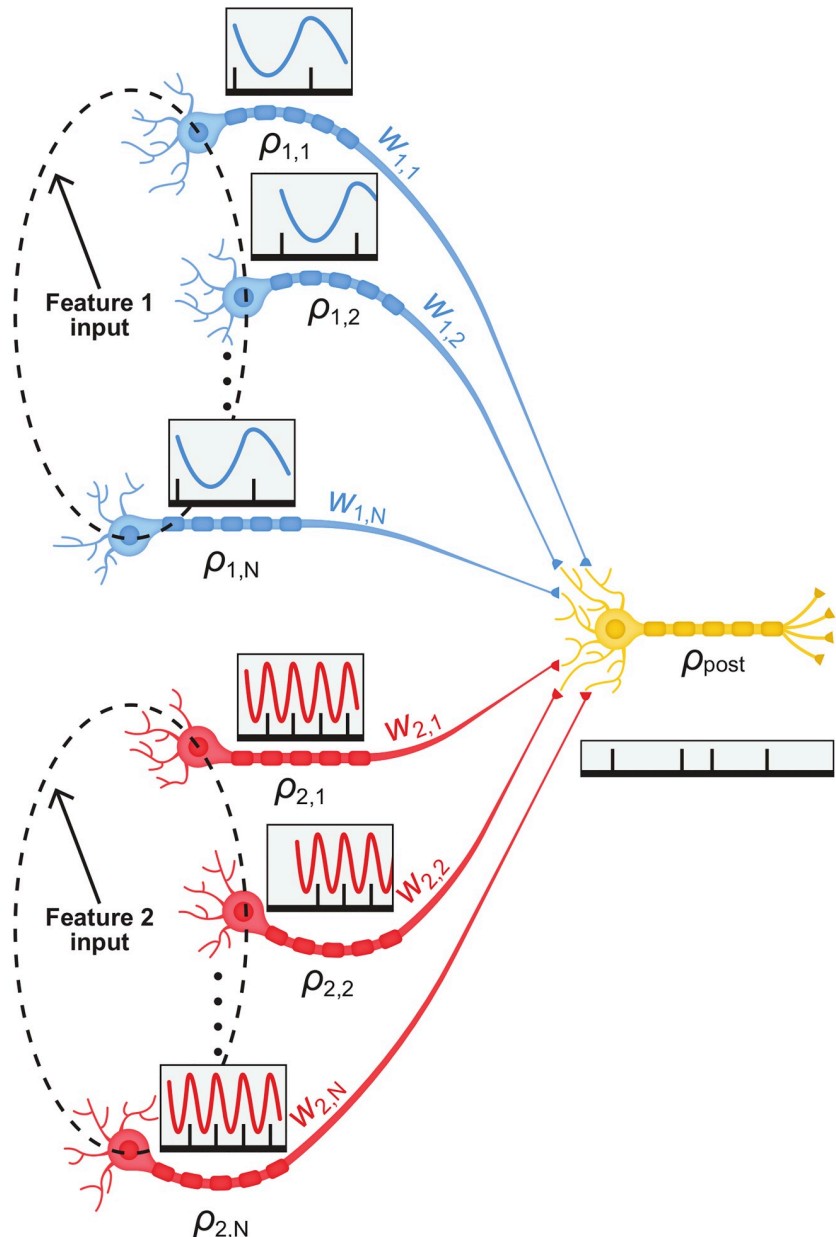

**Fig 1. Network architecture.** A schematic description of the network architecture showing two pre-synaptic populations, each oscillating at a different frequency. The output of these pre-synaptic neurons, serves as a feed-forward input to a single post-synaptic neuron.

for example typical values for the timescale for the stimulus are on the order of 1s, neural response will follow the stimulus within an order of 10ms and spike at about 10Hz.

As we are interested in studying multiplexing, we assume that the two populations are synapsing in a feed-forward manner onto the same downstream neuron.

## The downstream neuron model

Spike time correlations are the driving force of STDP dynamics [39, 45, 46, 50, 51]. Correlated pairs are more likely to affect the firing of the downstream (post-synaptic) neuron, and as a

result, to modify their synaptic connections [45, 46, 49]. To analyze the STDP dynamics we need a simplified model for the post-synaptic firing that will enable us to compute the pre-post cross-correlations, and in particular, their dependence on the synaptic weights.

Following past studies [35, 39, 48, 51–53], the post-synaptic neuron is modeled as a linear Poisson neuron with a characteristic delay $d > 0$. The mean firing rate of the post-synaptic neuron at time $t$, $r_{post}(t)$, is given by

$$r_{post}(t) = \frac{1}{N}\sum_{\eta=1}^{2}\sum_{k=1}^{N} w_{\eta,k}\rho_{\eta,k}(t - d),$$

(3)

where $w_{\eta,k}$ is the synaptic weight of the $k$th neuron of population $\eta$.

## Temporal correlations & order parameters

The utility of the linear neuron model is that the pre-post correlations are given as a linear combination of the correlations of the pre-synaptic populations. The cross-correlation between pre-synaptic neurons at time difference $\Delta t$ is given by:

$$\Gamma_{(\eta,j),(\xi,k)}(\Delta t) = \langle \rho_{\eta,j}(t)\rho_{\xi,k}(t + \Delta t)\rangle = \delta_{\eta\xi}\left(D^2(1 + \sigma^2)(1 + \frac{\gamma^2}{2}\cos[v_\eta\Delta t \right.$$

$$\left. + \phi_{\eta,j} - \phi_{\xi,k}]) + \delta_{jk}D\delta(\Delta t)\right) + D^2(1 - \delta_{\eta\xi}),$$

(4)

where $\delta_{\eta\xi} = 1$ when $\eta = \xi$ and 0 otherwise, is the Kronecker delta function.

The correlation between the $j$th neuron in population 1 and the post-synaptic neuron can therefore be written as

$$\Gamma_{(1,j),\ post}(\Delta t) = \frac{D}{N}\delta(\Delta t - d)w_{1,j} + D^2(1 + \sigma^2)\left(\bar{w}_1 + \frac{\gamma^2}{2}\tilde{w}_1\cos[v_1(\Delta t - d)\right.$$

$$\left. + \phi_{1,j} - \psi_1]\right) + D^2\bar{w}_2$$

(5)

in which $\delta(t)$ is the Dirac delta function, and the correlations are determined by global order parameters $\bar{w}$ and $\tilde{w}e^{i\psi}$, where $\bar{w}$ is the mean synaptic weight and $\tilde{w}e^{i\psi}$ is its first Fourier component. For $N \gg 1$ these parameters are defined as follows

$$\bar{w}_\eta(t) = \int_0^{2\pi} w_\eta(\phi, t)\frac{d\phi}{2\pi}$$

(6)

and

$$\tilde{w}_\eta(t)e^{i\psi_\eta} = \int_0^{2\pi} w_\eta(\phi, t)e^{i\phi}\frac{d\phi}{2\pi}.$$

(7)

The phase $\psi_\eta$ is determined by the condition that $\tilde{w}_\eta$ is real non-negative. Note that the coupling between the two populations is only expressed through the last term of the correlation function, Eq (5).

## The STDP rule

Following [46, 50, 51] we model the synaptic modification $\Delta w$ following either a pre- or post-synaptic spike as:

$$\Delta w = \lambda[f_+(w)K_+(\Delta t) - f_-(w)K_-(\Delta t)], \tag{8}$$

The STDP rule, Eq (8), is written as a sum of two processes: potentiation (+, increase in the synaptic weight) and depression (-, decrease). We further assume a separation of variables by writing each process as a product of a weight dependent function, $f_\pm(w)$, and a temporal kernel, $K_\pm(\Delta t)$. The term $\Delta t = t_{post} - t_{pre}$ is the time difference between pre- and post-synaptic spiking. Here we assumed, for simplicity, that all pairs of pre and post spike times contribute additively to the learning process via Eq (8). Note, however, that the temporal kernels of the STDP rule, $K_\pm(\Delta t)$ have a finite support. Here we normalized the kernels, $\int K_\pm(\Delta t)d\Delta t = 1$. The parameter $\lambda$ is the learning rate. It is assumed that the learning process is slower than the neuronal spiking activity and the timescale of changes in the external stimulus. Thus, the synaptic weights are relatively fixed on timescales characterizing changes in the external stimulus and the neural response. Here, we used the synaptic weight dependent functions of the form of [46]:

$$f_+(w) \quad = (1-w)^\mu \tag{9}$$

$$f_-(w) \quad = \alpha w^\mu, \tag{10}$$

where $\alpha > 0$ is the relative strength of depression and $\mu \in [0, 1]$ controls the non-linearity of the learning rule. The functions $f(w)_\pm$ ensure that the synaptic weights are confined to the region $w \in [0, 1]$. Gütig and colleagues [46] showed that the relevant parameter regime for the emergence of a non-trivial structure is $\alpha > 1$ and small $\mu$. Gütig and colleagues have also showed that the limit of $\mu = 0$, termed the additive model enhances the competitive nature of STDP dynamics, whereas the limit of $\mu = 1$, termed the linear model, greatly suppresses the competitive nature.

Empirical studies reported a large repertoire of temporal kernels for STDP rules [37, 38, 40–42, 44, 54–56]. Here we focus on two families of STDP rules: 1. A temporally asymmetric kernel [37, 41, 44, 55]. 2. A temporally symmetric kernel [38, 42, 44, 56].

For the temporally asymmetric kernel we use the exponential model, Fig 2a:

$$K_\pm(\Delta t) = \frac{e^{\mp\Delta t/\tau_\pm}}{\tau_\pm}\Theta(\pm\Delta t), \tag{11}$$

where $\Delta t = t_{post} - t_{pre}$, $\Theta(x)$ is the Heaviside function, and $\tau_\pm$ is the characteristic timescale of the potentiation (+) or depression (−). We assume that $\tau_- > \tau_+$ as typically reported.

For the temporally symmetric learning rule we use a difference of Gaussians model, Fig 2b:

$$K_\pm(\Delta t) = \frac{1}{\tau_\pm\sqrt{2\pi}}e^{-\frac{1}{2}\left(\frac{\Delta t}{\tau_\pm}\right)^2}, \tag{12}$$

where $\tau_\pm$ is the temporal width. In this case, the order of firing is not important; only the absolute time difference. We further assume, in both models, that $\tau_+ < \tau_-$, as is typically reported.

## STDP dynamics in the limit of slow learning

Due to noisy neuronal activities, the learning dynamics is stochastic. However, in the limit of a slow learning rate, $\lambda \to 0$, the fluctuations become negligible and one can obtain deterministic

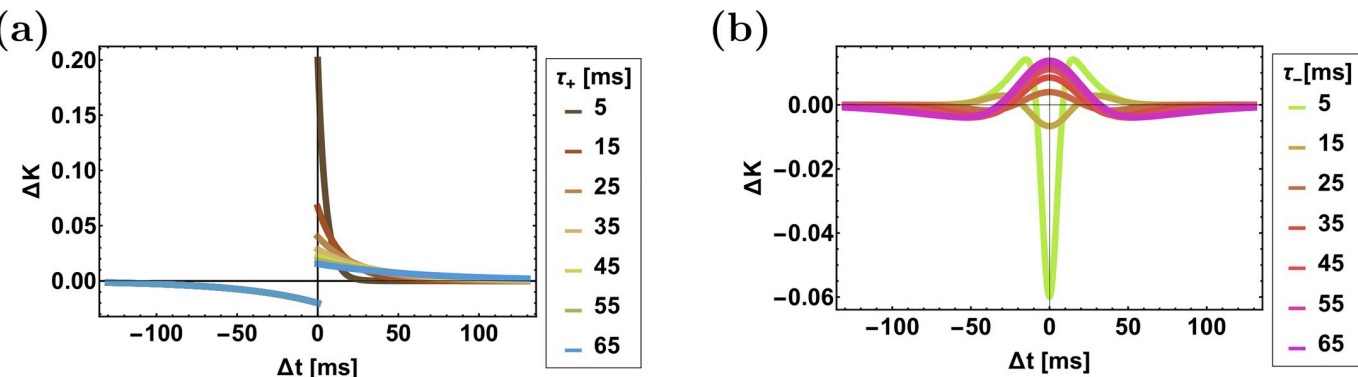

**Fig 2. The STDP rules.** The temporal kernels, $K(\Delta t) = K_+(\Delta t) - K_-(\Delta t)$, of the STDP rules are shown as a function of pre-post spike time difference, $\Delta t = t_{post} - t_{pre}$, for (**a**) The asymmetric learning rule, Eq (11) with $\tau_- = 50$ms. (**b**) The symmetric learning rule, Eq (12) with $\tau_+ = 20$ms. Different values of $\tau_+$ in (a) and $\tau_-$ in (b) are depicted by color as shown in the legend.

dynamic equations for the (mean) synaptic weights (see [50] for a detailed derivation)

$$\frac{\dot{w}_{\eta,j}(t)}{\lambda} = I^+_{(\eta,j)}(t) - I^-_{(\eta,j)}(t) \tag{13}$$

where $\eta = 1, 2$ and

$$I^\pm_{(\eta,j)}(t) = f_\pm(w_{\eta,j}(t)) \int_{-\infty}^{\infty} \Gamma_{(\eta,j),\ \text{post}}(\Delta) K_\pm(\Delta) d\Delta. \tag{14}$$

In Methods we derive the dynamics of the global order parameters using the correlation structure induced by the rhythmic activity, Eq (5). Note that in the dynamics of the order parameters, Eq (26), $\tilde{w}_2$ does not appear explicitly in the dynamics of $\tilde{w}_1$, and vice-versa. This results from the linearity of the post synaptic neuron model we chose, Eq (3).

**The homogeneous solution, Winner-take-all and multiplexing.**   Fig 3 shows three example results of simulating the STDP dynamics in the limit of slow learning, Eqs (25) and (26). Panels a & b show the dynamics of the synaptic weights of both populations, color coded by their preferred phase of firing. Panel c depicts the spectrum of the downstream (post-synaptic) neuron firing. Initially, as the synaptic weights are random, the input to the downstream neuron has almost no rhythmic component. In the example of Fig 3a–3c the synaptic weights of both populations converge to a homogeneous solution, in which all the synaptic weights from input neurons of different preferred phases and different populations are the same. As a result, the input to the downstream neuron has no rhythmic component and its activity shows no peak at any non-trivial frequency. The homogeneous solution is expected to be stable for large $\mu$ [46].

In the example of Fig 3d–3f rhythmic activity is transferred. As can be seen from Fig 3e, the homogeneous solution is not stable and the STDP dynamics causes the development of preference in the synaptic weights of population 2 to certain phases over the others. This allows the transmission of the rhythmic signal downstream. However, the STDP dynamics induces a winner-take-all competition and population 2 fully suppress population 1; hence, only one rhythmic signal is transmitted downstream and multiplexing is not enabled. The example of Fig 3g–3i depicts the desired scenario of multiplexing. In this case the homogeneous solution is unstable and both populations develop a phase preference; thus, enabling the transmission of rhythmic activity for both signals downstream.

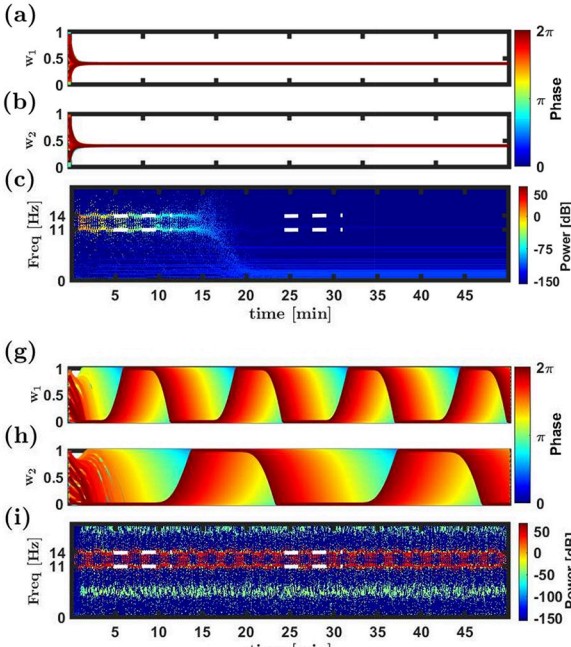

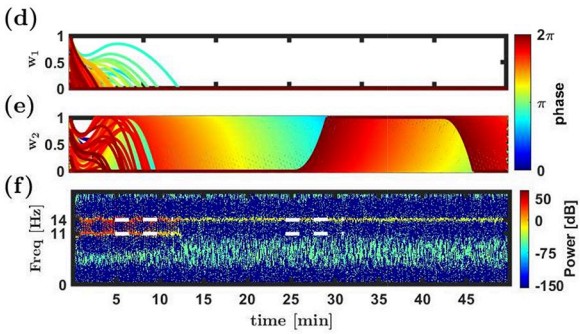

**Fig 3. Three examples: Homogeneous solution, winner-take-all and multiplexing.** Simulation results of the STDP dynamics in the limit of slow learning and a linear Poisson downstream neuron, Eq (13), are shown for three example cases. (**a**)-(**c**) The homogeneous solution, using $\mu = 0.1$ and $\alpha = 1.05$. (**d**)-(**f**) Winner-take-all competition, using $\mu = 0.001$ and $\alpha = 1.1$. (**g**)-(**i**) Multiplexing, using $\mu = 0.01$ and $\alpha = 1.05$. Panels a, b, d, e, g and h show the synaptic weights as a function of time. Each trace depicts the dynamics of a single synaptic weight. The synapses (traces) are differentiated by color according to the preferred phases of the corresponding pre-synaptic neurons, see legend. Panels c, f and i show the spectrogram of the downstream neuron. The horizontal dashed white lines depict the location of the rhythmic channels. In addition to the rhythmic channel (f) and rhythmic channels (i) one can identify sidebands in the spectrograms (f & i). Note that: 1) These additional bands are attenuated by about 100dB relative to the rhythmic channels. 2) They reflect the ongoing STDP dynamics; hence, they do not appear in c, and freezing the STDP abolishes these additional bands (results not shown). In all these examples the asymmetric learning rule, Eq (11), was used. The initial conditions of the synaptic weights were random. The synaptic weights at time zero were independent and identically distributed uniformly on [0, 1]. The parameters that were used in these examples are: $v_1 = 11$Hz, $v_2 = 14$Hz, $N = 120$, $\lambda = 0.001$, $\gamma = 1$, $D = 10$Hz, $\sigma = 0.8$, $d = 10$ms, $\tau_+ = 20$ms and $\tau_- = 50$ms.

The above examples differ in the parameters that define their respective STDP rules. Below we aim obtain insight as to the conditions that allow multiplexing. Eq (23) describes high dimensional coupled non-linear dynamics for the synaptic weights. Studying the development of rhythmic activity is, thus, not a trivial task. To this end, we take an indirect approach. We study the stability of the homogeneous solution, $w_{\eta,i} = w_\eta^*$ for all $i$, in which the rhythmic activity does not propagate downstream, Fig 3a–3c. Specifically, we investigate the conditions in which the homogeneous solution is unstable, and the STDP dynamics can evolve to a solution that has the capacity to transmit rhythmic information in both channels downstream. A complete derivation of the stability analysis can be found in Methods.

**The homogeneous solution.** The symmetry of the STDP dynamics, Eq (23), with respect to rotation guarantees the existence of a uniform solution where $w_{\eta,j}(t) = w_\eta^* \; \forall j \in \{1, \ldots N\}$ and $\tilde{w}_\eta(t) = 0$ with $\eta = 1, 2$. Solving the fixed point equation for the homogeneous solution yields

$$\frac{f_-(w^*)}{f_+(w^*)} = \frac{1 + X_+}{1 + X_-} \equiv \alpha_c, \tag{15}$$

where

$$X_\pm \equiv \frac{1}{(2 + \sigma^2)ND} K_\pm(d) \geq 0. \tag{16}$$

 

Due to the scaling of $X_{\pm}$ with $N$, $\alpha_c$ is not expected to be far from 1. From symmetry $w_1^* = w_2^* = w^*$:

$$w^* = \left(1 + \left(\frac{\alpha}{\alpha_c}\right)^{1/\mu}\right)^{-1}. \tag{17}$$

Substituting the homogeneous solution into the post-synaptic firing rate equation, Eq (3) yields

$$\langle \rho_{post} \rangle = 2Dw^*. \tag{18}$$

Thus, in the homogeneous solution, the post-synaptic neuron will fire at a constant rate in time and the rhythmic information will not be relayed downstream.

**Stability of the homogeneous solution.** Performing standard stability analysis, we consider small fluctuations around the homogeneous fixed point, $w_{\eta,j} = w^* + \delta w_{\eta,j}$, and expand to first order in the fluctuations:

$$\delta \dot{\boldsymbol{w}} = \lambda D^2 \boldsymbol{M} \delta \boldsymbol{w}, \tag{19}$$

where $\boldsymbol{M}$ is the stability matrix. Analysis of the stability matrix yields four prominent eigenvalues, see Methods. The first, $\bar{\lambda}_u$, represents fluctuations in the uniform direction, in which all the synapses are either potentiating or depressing together, is always stable Fig 4a. Furthermore, $\bar{\lambda}_u$ provides a stabilizing term in other modes of fluctuation. We distinguish two regimes according to the relative strength of depression, $\alpha$. For $\alpha < \alpha_c$, $\lim_{\mu \to 0+} \bar{\lambda}_u = -\infty$, and the uniform solution is expected to remain stable. For $\alpha > \alpha_c$, $\lim_{\mu \to 0+} \bar{\lambda}_u = 0$, and structure may

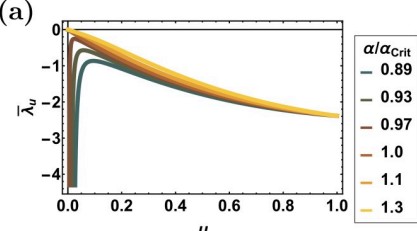

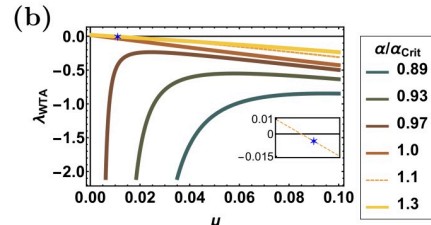

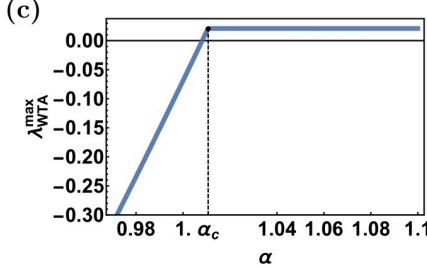

**Fig 4. The uniform and winner-take-all eigenvalues.** (a) The uniform eigenvalue, $\bar{\lambda}_u$, is shown as a function of $\mu$, for different values of $\alpha/\alpha_c$, depicted by color. (b) The competitive eigenvalue of the winner-take-all mode, $\lambda_{WTA}$, is shown as a function of $\mu$, for different values of $\alpha/\alpha_c$, depicted by color. (Inset) Enlarged section of the figure, showing the eigenvalue corresponding to the choice of parameters of Fig 8a–8d, depicted by a blue asterisk. (c) The maximal value of $\lambda_{WTA}$ in the interval $\mu \in [0, 1]$ is shown as a function of $\alpha$. Note that for $\alpha \geq \alpha_c$, $\lambda_{WTA}^{max}$ is obtained on the boundary $\mu = 0$. The eigenvalues were computed for the asymmetric STDP rule, Eq (11). Unless stated otherwise, the following parameters were used: $\sigma = 0.6$, $D = 10$Hz, $N = 120$, $\tau_- = 50$ms, $\tau_+ = 20$ms and $d = 10$ms.

 

emerge for sufficiently low values of $\mu$, Fig 4a. Thus, in order for the STDP dynamics to develop structure, whether winner-take-all of multiplexing, the relative strength of depression, $\alpha$, must be sufficiently high and $\mu$ has to be low.

The second eigenvalue, $\lambda_{\mathrm{WTA}}$, represents a winner-take-all like mode of fluctuations, in which synapses in one population suppress the other. Clearly, a winner-take-all competition will prevent multiplexing. Typically, for sufficiently high values of $\alpha$ and low values of $\mu$ the winner-take-all competitive mode will become unstable; hence, suppressing the option of multiplexing, Fig 4b and 4c (see Methods for complete analysis).

The last two prominent eigenvalues, $\tilde{\lambda}_{v_\eta}$ ($\eta = 1, 2$), are due to the rhythmic modes, see Eq (36) in Methods. Thus, $\tilde{\lambda}_{v_\eta}$, represents fluctuations in the synaptic weights of population $\eta$, in which a phase preference is developed, enabling the transmission of rhythmic activity in (angular) frequency $v_\eta$. Examining the rhythmic eigenvalues reveals that they are composed of a sum of two terms, see Methods. The first term is similar to $\lambda_{\mathrm{WTA}}$ and can become positive (unstable) for sufficiently large values of $\alpha$ and low values of $\mu$, and is largely independent of the temporal structure of the STDP rule. The second term depends on the rhythmic activity and the temporal structure of the STDP rule. It is this second term that can enable the rhythmic modes to develop while the competitive winner-take-all mode is suppressed.

Figs 5 and 6 show the dependence of the rhythmic eigenvalues on the various parameters that govern the STDP dynamics for the temporally asymmetric, Eq (11), and the temporally symmetric, Eq (12), learning rules, respectively. The temporal structure of the STDP rule, Eqs (11) and (12), as well as the delay, $d$, and the modulation depth, $\gamma$, determine the frequency

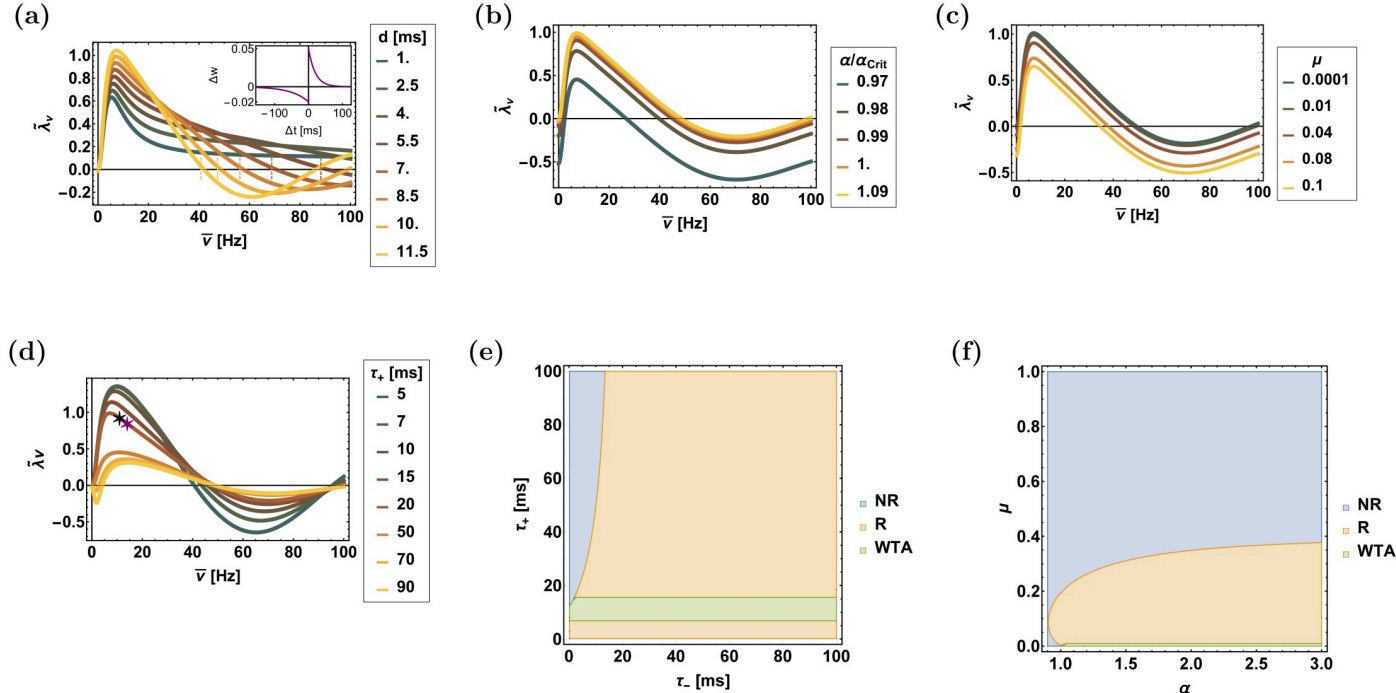

**Fig 5. The rhythmic eigenvalue, $\tilde{\lambda}_v$, for the asymmetric learning rule, Eq (11).** (a)-(d) The rhythmic eigenvalue, $\tilde{\lambda}_v$, is shown as a function of frequency, $\bar{v} = v/(2\pi)$, for different values of: the delay, $d$, in (a), relative strength of depression, $\alpha/\alpha_c$ in (b), $\mu$ in (c), and of the potentiation time constant, $\tau_+$, in (d)—as depicted by color. The black (11Hz) and purple (14Hz) stars in (d) show the eigenvalues with the parameters used in the simulations as shown in Fig 8a–8d. (e) and (f) The phase diagram of the system showing regions of different types of solutions in the plane of $[\tau_+, \tau_-]$ in (e) and the plane of $[\mu, \alpha]$ in (f), as determined by the signs of $\tilde{\lambda}_v$ and $\lambda_{\mathrm{WTA}}$. The abbreviations to the right of the panels are: NR—non rhythmic, R—rhythmic (multiplexing) and WTA—winner-take-all. Unless stated otherwise, the parameters used in this figure are: $\gamma = 1$, $\sigma = 0.6$, $D = 10$Hz, $N = 120$, $\tau_- = 50$ms, $\tau_+ = 20$ms, $\mu = 0.01$, $\alpha = 1.1$ and $d = 10$ms. In (d) $\mu = 0.011$.

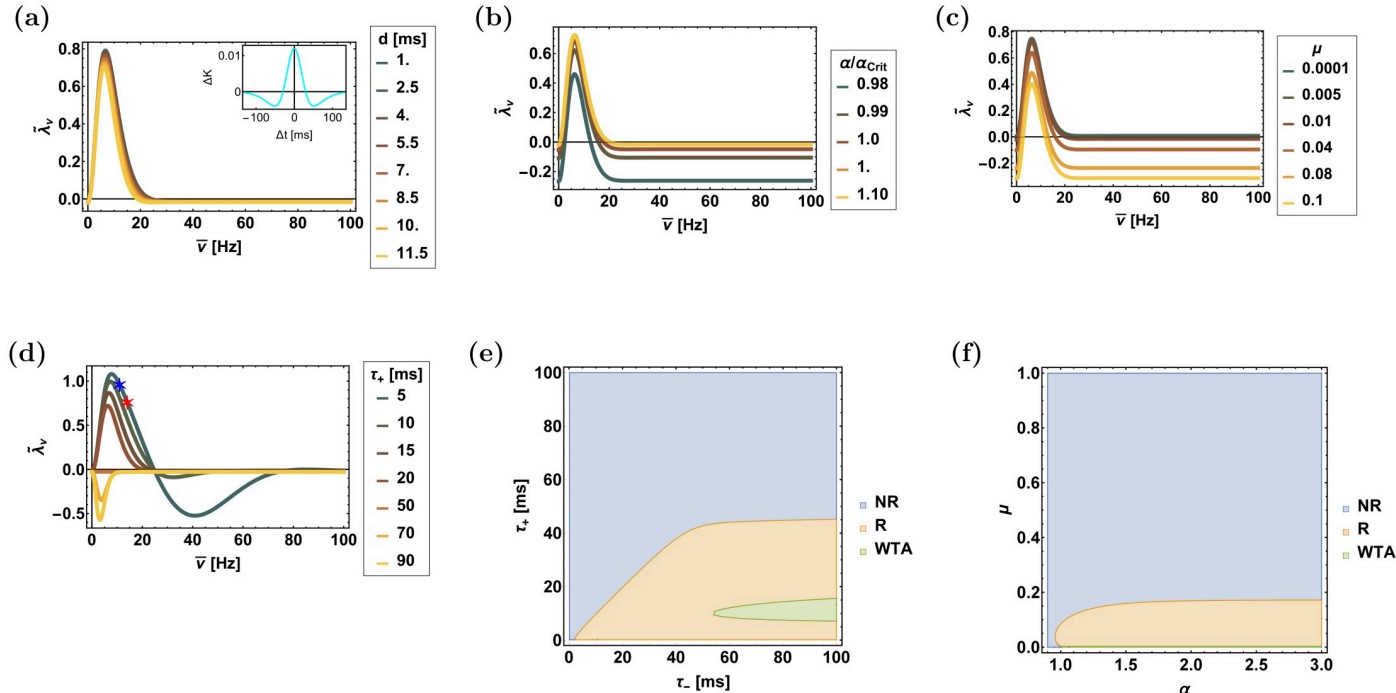

**Fig 6. The rhythmic eigenvalue, $\tilde{\lambda}_\nu$, for the symmetric learning rule, Eq (12).** (a)-(d) The rhythmic eigenvalue, $\tilde{\lambda}_\nu$, is shown as a function of frequency, $\bar{\nu} = \nu/(2\pi)$, for different values of: the delay, $d$, in (a), relative strength of depression, $\alpha/\alpha_c$ in (b), $\mu$ in (c), and of the potentiation time constant, $\tau_+$, in (d)—as depicted by color. The blue (11Hz) and red (14Hz) stars in (d) show the eigenvalues with the parameters used in the simulations as shown in S4a–S4d Fig. (e) and (f) The phase diagram of the system showing regions of different types of solutions in the plane of $[\tau_+, \tau_-]$ in (e) and the plane of $[\mu, \alpha]$ in (f), as determined by the signs of $\tilde{\lambda}_\nu$ and $\lambda_{WTA}$. The abbreviations to the right of the panels are: NR—non rhythmic, R—rhythmic (multiplexing) and WTA—winner-take-all. Unless stated otherwise, the parameters used in these figures are: $\gamma = 1$, $\sigma = 0.6$, $D = 10$Hz, $N = 120$, $\tau_- = 50$ms, $\tau_+ = 20$ms, $\mu = 0.01$, $\alpha = 1.1$ and $d = 10$ms. In (d) $\alpha = 1.05$ and $\mu = 0.011$, in (e) $\mu = 0.005$ was taken.

dependence of $\tilde{\lambda}_{\nu_\eta}$, whereas increasing $\alpha$ and decreasing $\mu$ shifts the curve up, increasing the range of unstable frequencies.

The above analysis provides intuition as to the required conditions for multiplexing. Essentially, one expects that if the competitive eigenvalue is stable, $\lambda_{WTA} < 0$, and both rhythmic eigenvalues are unstable, $\tilde{\lambda}_\nu > 0$, then the synaptic weights will evolve towards a state that will allow the transfer of both rhythms downstream. This is summarized in the phase diagram of the system, which depicts the different types of behaviour as a function of the parameters that characterize the STDP rule, panels e & f of Figs 5 and 6. Note, however, that the computation of these phase diagrams is based on local analysis of the homogeneous solution (the stability of $\lambda_{WTA}$, and instability of $\tilde{\lambda}_\nu$). To study the non-local behaviour, a numerical investigation is required.

Fig 7 depicts the results of a numerical simulation of the STDP dynamics with $\lambda_{WTA} \approx -0.003 < 0$, $\tilde{\lambda}_{\nu_1} \approx 0.93 > 0$, and $\tilde{\lambda}_{\nu_2} \approx 0.97 > 0$. Fig 7a and 7b show the dynamics of the synaptic weights. Initially, the synaptic weights were homogeneous (up to a small noise component, see caption) and no rhythm was transmitted downstream. However, through a process of spontaneous symmetry breaking both populations developed a phase preference; thus, enabling multiplexing.

Examining the dynamics of the order parameters, one can see how the rhythmic components, $\tilde{w}_1$ and $\tilde{w}_2$, evolve in time, rising from zero towards a fixed point value, Fig 7c. In

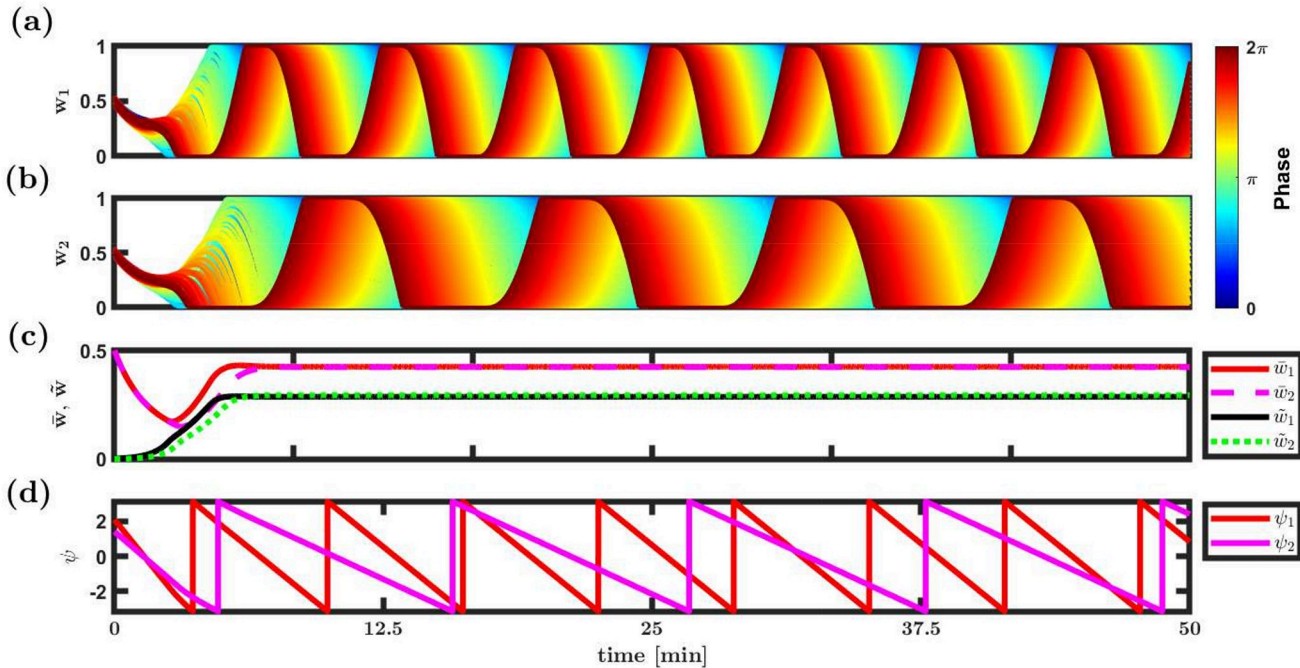

**Fig 7. Multiplexing as a limit cycle solution.** Simulation results of the STDP dynamics in the limit of slow learning and a linear Poisson downstream neuron, Eq (13). (**a**) and (**b**) The synaptic weights are shown as a function of time, for populations 1 and 2 in (a) and (b), respectively. Each trace depicts the dynamics of a single synaptic weight. The synapses (traces) are differentiated by color according to the preferred phases of the pre-synaptic neurons, see legend. (**c**) The dynamics of the order parameters: mean, $\bar{w}$, and the magnitude of the first Fourier component, $\tilde{w}$, are shown for populations 1 and 2, see legend. (**d**) The dynamics of the phases, $\psi_1$ and $\psi_2$, is shown in red and pink, respectively, as a function of time. The synaptic weights at tie zero were random, stochastically independent with identical uniform distribution on [0.45, 0.55]. The parameters used in this simulation are: $N = 120$, $\bar{v}_1 \equiv v_1/(2\pi) = 5$Hz, $\bar{v}_2 = 9$Hz, $\lambda = 0.001$, $\gamma = 1$, $\sigma = 0.6$, $D = 10$Hz, $N = 120$, $\tau_- = 50$ms, $\tau_+ = 20$ms, $\mu = 0.01$, $\alpha = 1.05$ and $d = 10$ms.

contrast with the order parameters, $\bar{w}$ and $\tilde{w}$, the synaptic weights themselves do not reach a fixed point, rather they remain dynamic. How can the order parameters remain fixed while the entire synaptic population is constantly changing? The solution to this puzzle is provided by examining the dynamics of $\psi_\eta$, see Eq (7), the phases of the rhythmic inputs, Fig 7d. As can be seen from the figure, the phases, $\psi_1$ and $\psi_2$, drift on the circle with constant velocities. Thus, the STDP dynamics converge to a limit cycle solution, in which the synaptic weights profile remains fixed—relative to to its phase, $w_\eta(\phi, t) = w_\eta(\phi - \psi_\eta(t))$, while the phases, themselves drift in time, $\psi_\eta(t) = \psi_\eta(0) + v_\eta t$. Qualitatively similar results can be obtained for the temporally symmetric STDP rule, Eq (12), see S3 Fig in Supporting information.

## Conductance based downstream neuron model

The above analysis relies on the choice of a linear neuron model, Eq (3), which resulted in the lack of explicit interaction term between the two rhythms $\tilde{w}_1$ and $\tilde{w}_2$, see Eq (26) in Methods. Non-linearity in the response of the downstream neuron to its inputs will generate interaction between the two rhythms. This interaction may increase the competition between the two rhythms that will prevent multiplexing. How robust are our results with respect to non-linear response of the downstream neuron? This issue is addressed below by studying the STDP dynamics in a conductance based Hodgkin-Huxley type model for the downstream neuron.

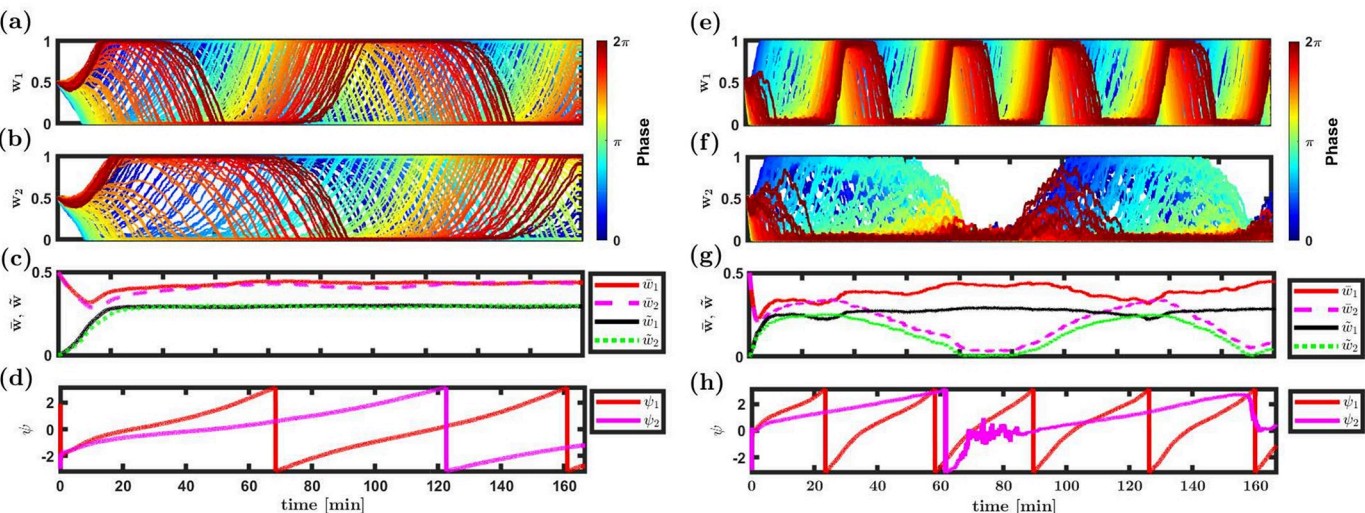

**Fig 8. STDP dynamics with a conductance based downstream neuron.** Results of two numerical simulation of STDP dynamics with a conductance based downstream neuron are presented: (**a**)-(**d**) using a downstream neuron with a linear f-I curve, and (**e**)-(**h**) using a downstream neuron with a non-linear f-I curve, see Details of numerical simulations in Methods. (a), (b), (e) and (f) The synaptic weights are shown as a function of time for population 1 (in a and e) and population 2 (in b and f). The different traces show the dynamics of different synapses colored by the preferred phase of their pre-synaptic neuron, see legend. (c) and (g) The dynamics of the order parameters: the mean, $\bar{w}$, and first Fourier component, $\tilde{w}$, are shown as a function of time for both populations, see legend. (d) and (h) The dynamics of the phases, $\psi_1$ and $\psi_2$, is shown as a function of time in red and pink, respectively. Here we used the temporally asymmetric STDP rule, Eq (11). Additional parameters are: $N_1 = N_2 = 120$, $\bar{v}_1 \equiv v_1/(2\pi) = 11$Hz, $\bar{v}_2 = 14$Hz, $\alpha = 1.1$ and $\mu = 0.011$. The learning rate $\lambda$ in the non-linear case is 5 times larger than in the linear case. For further details see Methods.

We used the conductance based model of Shriki et al. [57] for the downstream neuron. This choice was motivated by the ability to control the degree of non-linearity of the neuron's response to its inputs. Often the response of a neuron is quantified using an f-I curve, which maps the frequency (f) of the neuronal spiking response to a certain level of injected current (I). In the Shriki model [57], a strong transient potassium A-current yields a threshold linear f-I curve to a good approximation. Thus, by adjusting the strength of the A-current we can control the 'linearity' of the f-I curve of the downstream neuron.

Fig 8 presents the results of simulating the temporally asymmetric STDP dynamics with a conductance based downstream neuron. In Fig 8a–8d the downstream neuron is characterized by a strong A-current. In this regime, the f-I curve of the downstream neuron is well approximated by a threshold-linear function (see Fig 1 in [57]). Consequently, it is reasonable to expect that our analytical results will hold, in the limit of a slow learning rate. Indeed, even though the initial conditions of all the synapses are uniform, the uniform solution loses its stability, and a structure that shows phase preference emerges, Fig 8a and 8b. After about half an hour of simulation time, the STDP dynamics of each sub-population converges to an approximately periodic solution. The order parameters $\bar{w}$ and $\tilde{w}$ appear to converge to a fixed point, Fig 8c, while the phases $\psi_1$ and $\psi_2$ continue to drift with a relatively fixed velocity, Fig 8d. For the specific choice of parameters used in this simulation, the competitive winner-take-all eigenvalue is stable, see inset Fig 4b, whereas the rhythmic eigenvalue is unstable, see stars in Fig 5d.

Fig 8e–8h depict the STDP dynamics, for a non-linear downstream neuron. To this end we used the Shriki model [57] with no A-current. As can be seen from the figure, the system converges to a dynamical solution that shows some degree of similarity with the linear neuron (Fig 8a–8d). Specifically, the system relaxes to a dynamical solution that enables the transmission of

both rhythms, i.e., multiplexing. However, in the non-linear case the order parameters $\bar{w}$ and $\tilde{w}$ do not converge to a fixed point but fluctuate around some mean value. Qualitatively similar results can be obtained for the temporally symmetric STDP rule, Eq (12), see S4 Fig in Supporting information.

## Discussion

Rhythmic activity in the brain has fascinated and puzzled neuroscientists for more than a century. Nevertheless, the utility of rhythmic activity remains enigmatic. One explanation frequently put forward is that of the multiplexing of information. Our work provides some measure of support for this hypothesis from the theory of unsupervised learning.

We studied the computational implications of a microscopic learning rule, namely STDP, in the absence of a reward or a teacher signal. Previous work showed that STDP generates a strong winner-take-all like competition between subgroups of correlated neurons, thus effectively eliminating the possibility of multiplexing [46]. Our work demonstrates that rhythmic activity does not necessarily generate competition between different rhythmic signals. Moreover, we found that under a wide range of parameters STDP dynamics will develop spontaneously the capacity for multiplexing.

Not every learning rule, i.e., choice of parameters that describes the STDP update rule, will support multiplexing. This observation provides a natural test for our theory. Clearly, if multiplexing has evolved via a process of STDP, then the STDP rule must exhibit instability with respect to the rhythmic modes and stability against fluctuations in the winner-take-all direction. These constraints serve as basic predictions of our theory.

Khamechian and colleagues [28] suggested a model in which visual information from the ventral and dorsal pathways is transmitted to the prefrontal cortex by means of two well separated rhythms. Specifically, they proposed that the ventral pathway conveys information to the prefrontal cortex in the gamma band (40-70 Hz), whereas information from the dorsal pathway is relayed in the high gamma band (180-220 Hz). As the frequency bands are so segregated, does this necessarily imply two different STDP rules with different temporal characteristics are used, one for each stream? No. Interestingly, Khamechian and colleagues showed that for the dorsal stream to transmit information efficiently, the distribution of preferred phases must change from uniform to a non-uniform distribution. This is consistent with the homogeneous solution, implying that the rhythmic eigenvalue of the higher gamma is stable (as well as the competitive eigenvalue). Alternatively, the preferred phases of dorsal neurons can fluctuate from trial to trial, yielding a homogeneous correlation structure, on the timescale of plasticity. In this case, STDP will not develop a rhythmic solution regardless of the plasticity rule. Thus, to pursue our theory empirically, one has first to characterize the distribution of preferred phases of the upstream population, and especially, the stability of the relative phases over time. Secondly, one has to establish multiplexing: is there a subgroup of neurons that receives and responds-to both streams of information? Or is there a segregation of signals also at the level of the downstream population? Third, one has to characterize the STDP rule and compute its eigenvalues.

In our work we have made several simplifying assumptions to facilitate the analysis:

1. We limited the discussion to multiplexing of only two rhythms. Is our theory general? Can STDP support multiplexing of more than two rhythms?

2. We assumed the two populations are symmetric. This reduced the number of free parameters (e.g., $N$, $D$, $\sigma$ and so on). Will asymmetry generate a winner-take-all competition in which the 'stronger' population suppresses the other; thus, preventing multiplexing?

3. We further assumed symmetry within each population. Specifically, we assumed that the preferred phases of the neurons in each population are evenly spaced on the ring. However, if the preferred phases of the neurons are not distributed uniformly, then even a homogeneous solution will transmit the rhythm downstream. In that case, can multiplexing emerge in a trivial manner irrespective of the STDP rule?

STDP can support multiplexing of more than one rhythm, as illustrated in S1 Text in the section Supporting information that shows the multiplexing of three rhythms. Can STDP support the multiplexing of 10, 100, 1000 rhythms? Is there a capacity limit? We believe that the nature of rhythmic activity in the brain limits the number of channels that can be efficiently multiplexed. This is due to the fact the rhythmic activity in the brain is relatively wide band and efficient multiplexing requires the different signals to be well segregated in frequency. Consequently, we believe that if frequency division multiplexing is used in the brain, then the number of multiplexed signals is small.

Question number 2 is addressed in S2 Text where we show that fine tuning of a symmetry between the two populations is not required in order to obtain multiplexing. Regarding question number 3, while it is true that a non uniform phase distribution will make the transmission of rhythmic activity easier, it will not abolish the winner-take-all like competition between the two signals. Furthermore, by selectively potentiating certain phases and depressing others, STDP has the ability to amplify the transmitted rhythmic component. This issue and the question of transmitting information about the phase of the rhythmic activity are beyond the scope of the current work and will be addressed elsewhere.

In Luz & Shamir [35], due to the underlying U(1) symmetry, the system converged to a limit cycle solution. For a single population, the order parameter, $\tilde{w}e^{i\psi}$, will drift on the ring $|\mathbf{w}| = const$ with a constant velocity. Here, in the linear neuron model, in the limit of a slow learning one expects the system to converge to the product space of two limit cycles. As there is no reason to expect that the ratio of drift velocities of the two populations will be rational, the order parameters $\tilde{w}_1 e^{i\psi_1}$, $\tilde{w}_2 e^{i\psi_2}$ will, most likely, cover the torus uniformly. Nevertheless, the reduced dynamics of each population will exhibit a limit cycle. This intuition relies on the lack of interaction between $\tilde{w}^1$ and $\tilde{w}^2$ in the dynamics of the order parameters, Eq (26). Essentially, the interaction between the two populations is mediated solely via their mean component, $\bar{w}$. However, introducing non-linearity to the response properties of the post-synaptic neuron will induce an interaction between the modes. Similarly, for any finite learning rate, $\lambda \neq 0$, the rhythmic modes will not be orthogonal and consequently will be correlated.

A post-synaptic neuron with a non-linear f-I curve and a finite learning rate is expected to induce an interaction between the two populations. Consequently, for a finite learning rate the order parameters $\tilde{w}_1 e^{i\psi_1}$, $\tilde{w}_2 e^{i\psi_2}$ will not be confined to a torus and $\tilde{w}_\eta e^{i\psi_\eta}$ will fluctuate around (in contrast with on) the ring. Traces for this behaviour can be seen by the fact that drift velocity in the numerical simulations is not constant (compare Fig 8d and 8h) and the global order parameters $\bar{w}$ and $\tilde{w}$ do not converge to a fixed point, but remain to fluctuate around some mean value (compare Fig 8c and 8g). Thus, the system converges to a strange attractor around the torus. Nevertheless, in this example, the post-synaptic neuron responds to both rhythms; hence, multiplexing does not require a linear neuron.

Synaptic weights in the central nervous system are highly volatile and demonstrate high turnover rates as well as considerable size changes that correlate with the synaptic weight [58–65]. How can the brain retain functionality in the face of these considerable changes in synaptic connectivity? Our work demonstrates how functionality, in terms of retaining the ability to transmit downstream rhythmic information in several channels, can be retained even when

the entire synaptic population is modified throughout its entire dynamic range. Here, robustness of function is ensured by the dynamics of the global order parameters.

## Methods

### STDP dynamics of the order parameters

Using the correlation structure, Eqs (5) and (14) yields

$$
I^{\pm}_{(\eta,j)}(t) \equiv f_{\pm}(w_{\eta j}(t)) D^2 \bar{K}_{\pm} \Bigg( \bar{w}_{\eta}(t)(1+\sigma^2) + \bar{w}_{\xi} + \frac{\gamma^2}{2}(1+\sigma^2) \frac{\tilde{K}_{\pm}}{\bar{K}_{\pm}}
$$

$$
\tilde{w}_{\eta}(t) \cos[\phi_{\eta j} - \Omega^{\eta}_{\pm} - v_{\eta} d - \psi_{\eta}] + \frac{1}{ND\bar{K}_{\pm}} K_{\pm}(d) w_{\eta j} \Bigg),
$$

(20)

where $\bar{K}_{\pm}$ and $\tilde{K}_{\pm} e^{i\Omega^{\eta}_{\pm}}$ are the Fourier transforms of the STDP kernels

$$
\bar{K}_{\pm} = \int_{-\infty}^{\infty} K_{\pm}(\Delta) d\Delta,
$$

(21)

$$
\tilde{K}_{\pm} e^{i\Omega^{\eta}_{\pm}} = \int_{-\infty}^{\infty} K_{\pm}(\Delta) e^{-iv_{\eta}\Delta} d\Delta.
$$

(22)

Note that for our specific choice of kernels, $\bar{K}_{\pm} = 1$, by construction.

The dynamics of the synaptic weights can be written in terms of the order parameters, $\bar{w}$ and $\tilde{w}$ (see Eqs (6) and (7)). In the continuum limit, Eq (13) becomes

$$
\frac{\dot{w}_{\eta}(\phi, t)}{\lambda} = F_{\eta,d}(\phi, t) + \bar{w}_{\eta}(t) F_{\eta,0}(\phi, t)(1+\sigma^2) +
$$

$$
\tilde{w}_{\eta}(t) F_{\eta,1}(\phi, t) + \bar{w}_{\xi}(t) F_{\eta,0}(\phi, t),
$$

(23)

where

$$
F_{\eta,d}(\phi, t) = w_{\eta}(\phi, t) \frac{D}{N} \Bigg( f_{+}(w_{\eta}(\phi, t)) K_{+}(d) -
$$

$$
f_{-}(w_{\eta}(\phi, t)) K_{-}(d) \Bigg),
$$

$$
F_{\eta,0}(\phi, t) = D^2 \Bigg( \bar{K}_{+} f_{+}(w_{\eta}(\phi, t)) - \bar{K}_{-} f_{-}(w_{\eta}(\phi, t)) \Bigg),
$$

(24)

$$
F_{\eta,1}(\phi, t) = D^2 \frac{\gamma^2}{2}(1+\sigma^2) \Bigg( \tilde{K}_{+} f_{+}(w_{\eta}(\phi, t)) \cos[\phi - \Omega^{\eta}_{+} -
$$

$$
v_{\eta} d - \psi_{\eta}] - \tilde{K}_{-} f_{-}(w_{\eta}(\phi, t)) \cos[\phi - \Omega^{\eta}_{-}
$$

$$
- v_{\eta} d - \psi_{\eta}] \Bigg).
$$

Integrating Eq (23) over $\phi$ yields the dynamics of the order parameters

$$
\frac{\dot{\bar{w}}_{\eta}(t)}{\lambda} = \bar{F}_{\eta,d}(t) + \bar{w}_{\eta}(t) \bar{F}_{\eta,0}(t)(1+\sigma^2) +
$$

$$
\tilde{w}_{\eta}(t) \bar{F}_{\eta,1}(t) + \bar{w}_{\xi}(t) \bar{F}_{\eta,0}(t),
$$

(25)

$$\frac{1}{\lambda}\frac{d}{dt}\left(\tilde{w}_\eta(t)e^{i\psi_\eta}\right) = \frac{e^{i\psi_\eta}}{\lambda}\left(\dot{\tilde{w}}_\eta(t) + i\tilde{w}_\eta(t)\dot{\psi}_\eta(t)\right) =$$

$$\tilde{F}_{\eta,d}(t)e^{i\Phi_{\eta,d}} + \bar{w}_\eta(t)(1+\sigma^2)\tilde{F}_{\eta,0}(t)e^{i\Phi_{\eta,0}} + \tilde{w}_\eta\tilde{F}_{\eta,1}(t) \tag{26}$$

$$e^{i\Phi_{\eta,1}} + \bar{w}_\xi(t)\tilde{F}_{\eta,0}(t)e^{i\Phi_{\eta,0}},$$

where

$$\bar{F}_{\eta,x}(t) = \int_{-\pi}^{\pi} F_{\eta,x}(\phi,t)\frac{d\phi}{2\pi},$$

$$\tilde{F}_{\eta,x}(t)e^{i\Phi_{\eta,x}} = \int_{-\pi}^{\pi} e^{i\phi}F_{\eta,x}(\phi,t)\frac{d\phi}{2\pi}, \quad x = d, 0, 1. \tag{27}$$

Note, that in Eq (26), $\tilde{w}_2$ does not appear explicitly in the dynamics of $\tilde{w}_1$.

## Analysis of the stability matrix

Below we analyze the stability matrix with respect to fluctuations around the homogeneous solution, **M**, Eq (19). Using Eqs (13) and (20), the fluctuations can be written as

$$\delta\dot{w}_{\eta,j} = \delta I^+_{(\eta,j)} - \delta I^-_{(\eta,j)}. \tag{28}$$

with

$$\delta I^\pm_{(\eta,j)} = \frac{\partial f_\pm(w^*)}{\partial w_{\eta,j}}D^2(2+\sigma^2)(1+X_\pm)w^*\delta w_{\eta,j} + f_\pm D^2\left((1+\sigma^2)\delta\bar{w}_\eta + \delta\bar{w}_\xi + \right.$$

$$\left. \frac{\gamma^2}{2}(1+\sigma^2)\tilde{K}_\pm\cos[\phi_{\eta,j} - \Omega^\eta_\pm - v_\eta d - \psi_\eta]\delta\tilde{w}_\eta\right) + \frac{f_\pm(w^*)}{N}DK_\pm(d)\delta w_{\eta,j}. \tag{29}$$

Without loss of generality, taking $\eta = 1$, $\xi = 2$ yields

$$\delta\dot{w}_{1,j} = -\hat{g}_0\delta w_{1,j} - \Delta f(w^*)(\delta\bar{w}_1 + \delta\bar{w}_2) - \sigma^2\Delta f(w^*)\delta\bar{w}_1 + \gamma^2(1+\sigma^2)(f_+(w^*)$$

$$\tilde{K}_+(v_1)\cos[\phi_{1,j} - \Omega^1_+ - v_1 d - \psi_1] - f_-(w^*)\tilde{K}_-(v_1)\cos[\phi_{1,j} - \Omega^1_- - v_1 d - \psi_1]) \tag{30}$$

$$\delta\tilde{w}_1.$$

In the homogeneous fixed point, Eq (15):

$$\hat{g}_0 = (2+\sigma^2)\left(\alpha\mu(1+X_-)\frac{w^{*\mu}}{1-w^*} + f_+(w^*) - f_-(w^*)\right)$$

$$= g_0 - (2+\sigma^2)\Delta f(w^*), \tag{31}$$

where

$$g_0 \equiv \alpha\mu(2+\sigma^2)(1+X_-)\frac{w^{*\mu}}{1-w^*}$$

$$\Delta f(w) \equiv f_-(w) - f_+(w). \tag{32}$$

Studying Eq (30), the stability matrix, **M**, has four prominent eigenvalues. Two are in the subspace of the uniform directions of the two populations, and two are in directions of the of

the rhythmic modes. As the uniform modes of fluctuations, $\delta\bar{\boldsymbol{w}}^\top = (\delta\bar{w}_1, \ \delta\bar{w}_2)$, span an invariant subspace of the stability matrix, $\boldsymbol{M}$, we can study the restricted matrix, $\bar{\boldsymbol{M}}$, defined by:

$$\delta\dot{\bar{\boldsymbol{w}}} = \lambda D^2 \bar{\boldsymbol{M}} \delta\bar{\boldsymbol{w}}. \tag{33}$$

The matrix $\bar{\boldsymbol{M}}$ has two eigenvectors, $\boldsymbol{v}_\mathrm{u}^\top = (1, 1)$ and $\boldsymbol{v}_\mathrm{WTA}^\top = (1, -1)$, and the corresponding eigenvalues are

$$\bar{\lambda}_\mathrm{u} = -g_0 \tag{34}$$

$$\lambda_\mathrm{WTA} = \bar{\lambda}_\mathrm{u} + 2\Delta f(w^*), \tag{35}$$

The first eigenvector represents the uniform mode of fluctuations and its eigenvalue is always negative, Fig 4a; hence, the homogeneous solution is always stable with respect to uniform fluctuations. Furthermore, $\bar{\lambda}_\mathrm{u}$ serves as a stabilizing term in other modes of fluctuations. We distinguish two regimes: $\alpha < \alpha_c$ and $\alpha > \alpha_c$. For $\alpha < \alpha_c$, $\lim_{\mu\to 0+}\bar{\lambda}_\mathrm{u} = -\infty$, and the uniform solution is expected to remain stable. For $\alpha > \alpha_c$, $\lim_{\mu\to 0+}\bar{\lambda}_\mathrm{u} = 0$, and structure may emerge for sufficiently low values of $\mu$, Fig 4a.

The second eigenvalue represents a winner-take-all like mode of fluctuations, in which synapses in one population suppress the other, and will prevent multiplexing. For $\alpha > \alpha_c$, $\lim_{\mu\to 0+}\lambda_\mathrm{WTA} = 2(\alpha_c - 1)$. For the temporally asymmetric learning rule, Eq (11), $X_- = 0$; consequently $\alpha_c > 1$. In the temporally symmetric difference of Gaussians STDP model, Eq (12), $\alpha_c > 1$ if and only if $\tau_+ < \tau_-$, which is the typical case. For $\alpha < \alpha_c$ in the limit of small $\mu$ the divergence of $\bar{\lambda}_\mathrm{u}$ stabilizes fluctuations in this mode. In this case ($\alpha < \alpha_c$), $\lambda_\mathrm{WTA}$ reaches its maximum at an intermediate value of $\mu \in (0, 1)$, Fig 4b. For a small range of $\alpha < \alpha_c$, $\alpha \approx \alpha_c$ this maximum can be positive, see Fig 4c. Fig 4b depicts $\lambda_\mathrm{WTA}$ as a function of $\mu$ for different values of $\alpha$, shown by color. Note that $\lambda_\mathrm{WTA}$ depends on the temporal structure of the STDP rule solely via the value of $\alpha_c$ and $X_-$; however, its sign is independent of $X_-$. As can seen in the figure, for $\alpha > \alpha_c > 1$, $\lambda_\mathrm{WTA}$ is a decreasing function of $\mu$ and the homogeneous solution loses stability in the competitive winner-take-all direction in the limit of small $\mu$. Note that the competitive eigenvalue is not identical in the asymmetric and symmetric rules due to the fact that $X_-^\mathrm{Symmetric} \neq X_-^\mathrm{Asymmetric} = 0$. However, as $X_-^\mathrm{Symmetric} \sim 10^{-3}$, $\lambda_\mathrm{WTA}$ behaves qualitatively the same in both types of STDP rules.

The rhythmic modes are eigenvectors of the stability matrix $\boldsymbol{M}$ with eigenvalues

$$\tilde{\lambda}_{v_\eta} = \bar{\lambda}_\mathrm{u} + (2 + \sigma^2)\Delta f(w^*) + \gamma^2(1 + \sigma^2)f_+(w^*)\tilde{Q} \tag{36}$$

$$\tilde{Q} = \tilde{K}_+(v_\eta)\cos[\Omega_+^\eta + v_\eta d] - \alpha_c\tilde{K}_-(v_\eta)\cos[\Omega_-^\eta + v_\eta d], \ \ (\eta = 1, 2). \tag{37}$$

The first two terms in the right hand side of Eq (36) contain the stabilizing term $\bar{\lambda}_\mathrm{u} \leq 0$, and their dependence on $\alpha$ and $\mu$ is similar to that of $\lambda_\mathrm{WTA}$; compare with Eq (35). The last term depends on the real part of the Fourier transform of the delayed STDP rule at the specific frequency of oscillations, $v_\eta$. This last term can destabilize the system in a direction that will enable the propagation of rhythmic activity downstream while keeping the competitive WTA mode stable depending on the interplay between the rhythmic activity and the temporal structure of the STDP rule. For $\tilde{Q} > 0$, $\tilde{\lambda}_{v_\eta}$ is an increasing function of the modulation to the mean ratio $\gamma$. If in addition $\alpha > \alpha_c > 1$ then $\tilde{\lambda}_{v_\eta}$ is an increasing function of $\sigma$ as well.

In the low frequency limit, $\lim_{v \to 0} \tilde{Q} = 1 - \alpha_c$, and depends on the characteristic timescales of $\tau_+$, $\tau_-$, and $d$ only via $\alpha_c$. For large frequencies $\lim_{v \to \infty} \tilde{Q} = 0$. In this limit the STDP dynamics loses its sensitivity to the rhythmic activity. Consequently, the resultant modulation of the synaptic weights profile, $\tilde{w}$, will become negligible; hence, effectively rhythmic information will not propagate downstream even if the rhythmic eigenvalue is unstable [35]. Thus, the intermediate frequency region is the most relevant for multiplexing.

In the case of the temporally symmetric kernel, Eq (12), the value of $\tilde{Q}$ is given by

$$\tilde{Q} = \cos[v_\eta d]\left( e^{-\frac{1}{2}(v_\eta \tau_+)^2} - \alpha_c e^{-\frac{1}{2}(v_\eta \tau_-)^2} \right),\tag{38}$$

where $\Omega_\pm^\eta = 0$. Fig 6a shows the rhythmic eigenvalue, $\tilde{\lambda}_v$, as a function of the oscillation frequency, $\bar{v} \equiv v/(2\pi)$, for different values of the delay, $d$ as depicted by color. Since typically, $\tau_+ < \tau_-$, for finite $v$, $\tilde{K}_+(v) > \tilde{K}_-(v)$. Consequently, $\tilde{Q}$ will be dominated by the potentiation term, $\cos[v_\eta d]e^{-\frac{1}{2}(v_\eta \tau_+)^2}$, except for the very low frequency range of $v \lesssim 1/\tau_-$. Typical values for the delay, $d$, are 1-10$ms$, whereas typical values for the characteristic timescales for the STDP, $\tau_\pm$, are tens of $ms$. As a result, the specific value of the delay, $d$, does not affect the rhythmic eigenvalue much, and the system becomes unstable in the rhythmic direction for $1/\tau_- \lesssim v \lesssim 1/\tau_+$.

Increasing the relative strength of the depression, $\alpha$, strengthens the stabilizing term $\bar{\lambda}_u$; however, $\Delta f(w^*)$ scales approximately linearly with $\alpha$ such that the rhythmic eigenvalue is elevated, causing the frequency range in which $\tilde{\lambda}_v > 0$ to widen, Fig 6b. Similarly, for $\alpha > \alpha_c > 1$, increasing $\mu$ strengthens $\bar{\lambda}_u$ and reduces the frequency range in which $\tilde{\lambda}_v > 0$, Fig 6c. Decreasing the characteristic timescale of potentiation, $\tau_+$ increases the frequency region with an unstable rhythmic eigenvalue; however, when $\tau_+$ becomes comparable to the delay, $d$, the oscillatory nature of $\tilde{\lambda}_v$ in $v$ is revealed, Fig 6d.

In the case of the temporally asymmetric kernel, Eq (11), the value of $\tilde{Q}$ is given by

$$\tilde{Q} = \frac{\cos[\Omega_+^\eta + v_\eta d]}{\sqrt{1 + (v_\eta \tau_+)^2}} - \alpha_c \frac{\cos[\Omega_-^\eta + v_\eta d]}{\sqrt{1 + (v_\eta \tau_-)^2}},\tag{39}$$

with $\Omega_\pm^\eta = \mp \arctan(v_\eta \tau_\pm)$. The main difference between the temporally symmetric and the asymmetric rules is that due to the discontinuity of the asymmetric STDP kernel, $\tilde{K}_\pm$ decay algebraically rather than exponentially fast with $v$. As a result, the phase $\cos[\Omega_\pm^\eta + v_\eta d]$, plays a more central role in controlling the stability of the rhythmic eigenvalue. As above, since typically, $\tau_+ < \tau_-$, then $\tilde{K}_+(v) > \tilde{K}_-(v)$. Fig 5a shows the rhythmic eigenvalue, $\tilde{\lambda}_v$, for different values of $d$. The dashed vertical lines depict the frequencies at which the potentiation term changes its sign, $\Omega_+^\eta + v^* d = \pi/2$. As can be seen from the figure, for this choice of parameters the upper cutoff of the central frequency range in which the rhythmic eigenvalue is unstable is dominated by $v^*$, which is governed by the delay.

The effects of parameters $\alpha$ and $\mu$ show similar trends as for the symmetric STDP rule. Specifically, increasing $\mu$ or decreasing $\alpha$, in general, shrinks the region in which fluctuations in the rhythmic direction are unstable, Fig 5b and 5c. Increasing the characteristic timescale of potentiation, $\tau_+$ beyond that of the depression, makes the depression term, $\tilde{K}_-(v_\eta)\cos[\Omega_-^\eta + v_\eta d]$, more dominant, Fig 5d. In this case the lower frequency cutoff will be dominated by the change of sign in the depression term; i.e., by the angular frequency $v^*$, such that $\Omega_-^\eta + v^* d = \pi/2$.

## Details of the numerical simulations

**The conductance based model of the downstream neuron.**   We used the conductance based model of Shriki et al. [57]. The model is fully defined and studied in [57]. Here, we briefly describe the model dynamics, as used in our numerical simulations. The dynamics of the membrane potential is given by:

$$C_m \dot{V} = -I_L - I_{Na} - I_k - I_A + \sum_\eta g_\eta (E_\eta - V), \tag{40}$$

where the leak current is given by $I_L = g_L(V - E_L)$. $I_{Na}$ and $I_k$ are the sodium and potassium currents, respectively, and are given by $I_{Na} = \bar{g}_{Na} m_\infty^3 h(V - E_{Na})$ and $I_k = \bar{g}_k n^4 h(V - E_k)$. The relaxation equations of the the gating variables $x = h$, $n$ are $dx/dt = (x_\infty - x)/\tau_x$. The time independent functions $x_\infty = h_\infty$, $n_\infty$, $m_\infty$ and $\tau_x$ are: $x_\infty = \alpha_x/(\alpha_x + \beta_x)$ and $\tau_x = 0.1/(\alpha_x + \beta_x)$, with $\alpha_m = -0.1(V + 30)/(\exp(-0.1(V + 30)) - 1)$, $\beta_m = 4 \exp(-(V + 55)/18)$, $\alpha_h = 0.07 \exp(-(V + 44)/20)$, $\beta_h = 1/(\exp(-0.1(V + 14)) + 1)$, $\alpha_n = -0.01(V + 34)/(\exp(-0.1(V + 34)) - 1)$ and $\beta_n = 0.125 \exp(-(V + 44)/80)$.

The A-current, $I_A$, that linearizes the f-I curve is given by $I_A = \bar{g}_A a_\infty^3 b(V - E_k)$, where $a_\infty = 1/(\exp(-(V + 50)/20) + 1)$ and $db/dt = (b_\infty - b)/\tau_A$. The time independent function of $b_\infty$ is $b_\infty = 1/(\exp((V + 80)/6) + 1)$ with the voltage independent time constant $\tau_A$.

The term $g_\eta$ is the total conductance of the pre-synaptic population $\eta$ and can be written as follows

$$g_\eta(t) = g_\eta^0 \sum_{j=1}^{N_\eta} \left( w_j^\eta(t) \sum_s \frac{[t - t_j^s]_+}{\tau_\eta} e^{-(t - t_j^s)/\tau_\eta} \right). \tag{41}$$

Here, $N_\eta$ is the number of neurons in population $\eta$, $w_j^\eta(t)$ is the synaptic weight of the $j$th neuron from population $\eta$ and $[y]_+ = max(0, y)$. The $s$ spike of the $j$th neuron is denoted by $t_j^s$. We used $g_\eta^0 = g_\eta^R S_x$ with $S_\eta = 1000/N_\eta$, $\tau_\eta = 5ms$ and $g_\eta^R = 900nS/cm^2$ (see [35, 46]).

The membrane capacity is $c_m = 0.1\mu F/cm^2$. The sodium, potassium and leak conductances are $\bar{g}_{Na} = 100mS/cm^2$, $\bar{g}_k = 40mS/cm^2$ and $g_L = 0.05mS/cm^2$. For the conductance based downstream neuron model with a *linear f-I curve* we used the following parameters: $g_A = 20mS/cm^2$ and $\tau_A = 20ms$. For the conductance based downstream neuron with a *non-linear f-I curve* we took $g_A = 0$. The reversal potentials of the ionic and synaptic currents are $E_{Na} = 55mV$, $E_K = -80mV$, $E_L = -65mV$, $E_\eta = E_{exc} = 0mV$.

**Modeling pre-synaptic activity.**   Pre-synaptic activities were modeled by independent inhomogeneous Poisson processes, with time dependent mean firing rate given by Eq (1), with $\gamma = 1$. Every second of simulation time $D_1$ and $D_2$ were independently sampled from a uniform distribution with a minimum of 7Hz and a maximum of 13Hz, $D = (7 + \mathbf{U}(0, 6))$Hz. Each pre-synaptic neuron, spiked according to an approximated Bernoulli process, with a probability of $p \approx r\Delta t$, where $r$ is the mean firing rate (Eq (1)) and $\Delta t = 1ms$. The number of pre-synaptic neurons in each population was $N = 120$.

**STDP.**   The learning rate of the simulations with a linear f-I curve is $\lambda = 0.01$, Fig 8a–8d and S4a–S4d Fig. In the non-linear cases the learning rate is $\lambda = 0.05$, Fig 8e–8h and S4e–S4h Fig. In both the linear and non-linear cases we used $\mu = 0.011$. In order to update the synaptic weights we relied on the separation of time scales between the synaptic dynamics of Eq (40); hence, the synaptic weights were updated every 1s of simulation.

- **Asymmetric learning rule**: The ratio of depression to potentiation is $\alpha = 1.1$ and the characteristic decay times were chosen to be $\tau_+ = 20ms$ and $\tau_- = 50ms$.

- **Symmetric learning rule**: Here, we chose $\alpha = 1.05$, furthermore, based on our analysis, we chose the ratio of decay times to be $\sim 10$, $\tau_+ = 5$ms, $\tau_- = 50$ms.

  Initial conditions for all neurons were uniform; i.e., $w_\eta(\phi, t = 0) = 0.5$, $\eta = 1, 2$.

## Supporting information

**S1 Text. Multiplexing three signals.**
(PDF)

**S2 Text. Multiplexing asymmetric signals.**
(PDF)

**S1 Fig. Multiplexing three signals for the temporally asymmetric STDP rule.** Simulation results of the STDP dynamics in the limit of slow learning and a linear Poisson downstream neuron, Eq (13), with three input signals. (**a**), (**b**) and (**c**) The synaptic weights are shown as a function of time, for populations 1,2 and 3, respectively. Each trace depicts the dynamics of a single synaptic weight. The synapses (traces) are differentiated by color according to the pre-ferred phases of thier pre-synaptic neurons, see legend. The initial conditions of the synaptic weights were random with uniform distribution on the interval [0, 1]. The parameters used in this simulation are: $N = 120$, $\lambda = 0.001$, $\bar{v}_1 \equiv v_1/(2\pi) = 7$Hz, $\bar{v}_2 = 11$Hz, $\bar{v}_3 = 15$Hz, $D = 10$Hz, $\sigma = 0.81$, $\gamma = 0.9$. We simulated the temporally asymmetric STDP rule, Eq (11), with $\tau_- = 50$ms, $\tau_+ = 5$ms, $\mu = 0.01$, $\alpha = 1.01$. The delay of the downstream neuron was $d = 10$ms.
(TIF)

**S2 Fig. Multiplexing asymmetric signals for the temporally asymmetric STDP rule.** Simula-tion results of the STDP dynamics in the limit of slow learning and a linear Poisson down-stream neuron, Eq (13), for two asymmetric signals. (**a**) and (**b**) The synaptic weights are shown as a function of time, for populations 1 and 2, respectively. Each trace depicts the dynamics of a single synaptic weight. The synapses (traces) are differentiated by color according to the pre-ferred phases of their pre-synaptic neurons, see legend. The initial conditions of the synaptic weights were random with uniform distribution on the interval [0, 1]. The parameters used in this simulation are $N = 120$, $\lambda = 0.001$, $\bar{v}_1 \equiv v_1/(2\pi) = 9$Hz, $\bar{v}_2 = 15$Hz, $A_1 = A_2 = 10$Hz, $D_1 = 15$Hz, $D_2 = 10$Hz. We simulated the temporally asymmetric STDP rule, Eq (11), with: $\tau_- = 50$ms, $\tau_+ = 20$ms, $\mu = 0.01$, $\alpha = 1.05$. The delay of the downstream neuron was $d = 10$ms.
(TIF)

**S3 Fig. Multiplexing as a limit cycle solution, symmetric learning rule.** Simulation results of the STDP dynamics in the limit of slow learning and a linear Poisson downstream neuron, Eq (13). (**a**) and (**b**) The synaptic weights are shown as a function of time, for populations 1 and 2 in (a) and (b), respectively. Each trace depicts the dynamics of a single synaptic weight. The synapses (traces) are differentiated by color according to the preferred phases of their pre-syn-aptic neurons, see legend. (**c**) The dynamics of the order parameters: mean, $\bar{w}$, and the magni-tude of the first Fourier component, $\tilde{w}$, are shown for populations 1 and 2, see legend. (**d**) The dynamics of the phases, $\psi_1$ and $\psi_2$, is shown in red and pink, respectively, as a function of time. The parameters used in this simulation are: $N = 120$, $\bar{v}_1 \equiv v_1/(2\pi) = 5$Hz, $\bar{v}_2 = 14$Hz, $\lambda = 0.001$, $\gamma = 1$, $\sigma = 0.6$, $D = 10$Hz, $N = 120$, $\tau_- = 50$ms, $\tau_+ = 5$ms, $\mu = 0.001$, $\alpha = 1.05$ and $d = 10$ms.
(TIF)

**S4 Fig. STDP dynamics with conductance based downstream neuron for the temporally symmetric STDP rule.** Results of two numerical simulation of STDP dynamics with a conduc-tance based downstream neuron are presented: (**a**)-(**d**) using a downstream neuron with a

linear f-I curve, and (**e**)-(**h**) using a downstream neuron with a non-linear f-I curve, see Details of numerical simulations Methods. (a), (b), (e) and (f) The synaptic weights are shown as a function of time for population 1 (in a and e) and population 2 (in b and f). The different traces show the dynamics of different synapses colored by the preferred phase of the pre-synaptic neuron, see legend. (c) and (g) The dynamics of the order parameters: the mean, $\bar{w}$, and first Fourier component, $\tilde{w}$, are shown as a function of time for both populations, see legend. (d) and (h) The dynamics of the phases, $\psi_1$ and $\psi_2$, is shown as a function of time in red and pink, respectively. Here we used the temporally symmetric STDP rule, Eq (12). Additional parameters are: $\bar{v}_1 \equiv v_1/(2\pi) = 11$Hz, $\bar{v}_2 = 14$Hz, $\alpha = 1.05$ and $\mu = 0.011$. The learning rate $\lambda$ in the non-linear case is 5 times larger than in the linear case. Further details of the numerical simulations appear in Methods.
(TIF)

## Author Contributions

**Conceptualization:** Maoz Shamir.

**Investigation:** Nimrod Sherf, Maoz Shamir.

**Software:** Nimrod Sherf.

**Writing – original draft:** Nimrod Sherf, Maoz Shamir.

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
