## [Decision Letter · Decision Letter 0]

25 Feb 2020

Dear Mr. Sherf,

Thank you very much for submitting your manuscript "Multiplexing rhythmic information by spike timing dependent plasticity" for consideration at PLOS Computational Biology.

As with all papers reviewed by the journal, your manuscript was reviewed by members of the editorial board and by several independent reviewers. In light of the reviews (below this email), we would like to invite the resubmission of a significantly-revised version that takes into account the reviewers' comments.

Note that both reviewers express confusion about precisely what is meant by multiplexing in this work, so clearly more attention is needed to clarifying and illustrating this point.  Similarly, there is agreement that the calculations could be presented more effectively, with the likely best option being to move the core calculations to Methods and make use of supplementary material opportunities to provide more details and explanations.

We cannot make any decision about publication until we have seen the revised manuscript and your response to the reviewers' comments. Your revised manuscript is also likely to be sent to reviewers for further evaluation.

Sincerely,

Jonathan Rubin

Associate Editor

PLOS Computational Biology

Lyle Graham

Deputy Editor

PLOS Computational Biology

Reviewer's Responses to Questions

**Comments to the Authors:**

Reviewer #1: Multiplexing is a method by which multiple signals are combined into one transmittable signal. There are various methods for multiplexing. For example, multiplexing can be achieved using alternating time slots, a method known as time-division multiplexing. In this study, the authors demonstrate that the spike-timing-dependent-plasticity, rather than supporting a Winner-Take-All dynamics, can underlie multiplexing. I think that they mean something like time-division multiplexing but I could be wrong.

I find the results of this paper both interesting and novel and I strongly recommend that it would be published in PCB. However, I think that the presentation of the results can be improved, and that additional analysis would be useful, as discussed below.

Major comments:

1. Three models are described: a linear Poisson model, a conductance-based model whose f-I curve is approximately threshold linear and a conductance-based model whose f-I curve substantially deviates from threshold linearity. For the linear model, it is demonstrated that for a range of parameters, the homogeneous fixed point is unstable in the rhythmic direction. For the two conductance-based models, numeric simulations demonstrate dynamic instability of the synaptic weights and their convergence to something that looks like a limit cycle.

a) In order for the reader to be able to compare the linear and conductance-based models, the authors should plot the dynamics of the weights in the linear model. As discussed in lines 239-240, the stability analysis is local, and while providing useful insight it would be useful to see the results of a simulation of the (deterministic approximation) of the linear model.

b) In order to demonstrate multiplexing, the authors should show (using their existing simulations) the temporal dynamics of the total input to the output neuron. This will also clarify the kind of multiplexing that operates in this problem, and what information about the two incoming signals is maintained over time.

c) Most of the paper is devoted to the stability analysis of the fixed-point. This part is pretty technical and obscures the results. I strongly recommend that this analysis will be moved to the Methods section, leaving the space in the Results section to explaining only the outline of the stability analysis and the intuition behind the instability of the homogeneous fixed point and the emergence of phase oscillations. It is also recommended to add one or two parameter-space plots, which depict the dependence of the qualitatively different behaviors of the linear model (estimated from the fixed-point analysis) on the different parameters.

Additional comments:

1. The abstract is too technical for a journal like PCB. It should be simplified and clarified.

2. The authors discuss too briefly the possibility of multiplexing in the brain and the relevance of their work to specific experiments. I recommend discussing these issues in more details in the Introduction and Discussion sections. Specifically, I wonder if the results presented in the current manuscript can be related to recently-described olfactory (https://www.nature.com/articles/nrn3668 ) and auditory (https://www.nature.com/articles/s41467-018-05121-8) multiplexing.

3. Correct me if I am wrong but if I understand it correctly, there are three time scales in the problem: t1=1/nu, the time scale of the oscillations, t2, the time-scale of the changes in the intensity D and t3, the time-scale of synaptic changes (this is in addition to the time-scale of the STDP rule). The assumption is that t1<<t2<<3. be="" clarified="" in="" should="" text="" the="" this="">4. What would happen if the number of inputs N associated with each of the signals is not equal for the two signals, or alternatively, if the magnitudes of the two signals are not equal on average?

5. What would happen if the number of signals is larger than 2? In other words, what is the "capacity" of the neuron for multiplexing? To clarify, for both this and the previous point, an analytical result is not requested and it would be sufficient to provide a result that is based on a small number of numeric simulations of the linear model for a particular set of parameters.

6. Demonstrating that multiplexing is not restricted to the linear case using a non-linear conductance-based model is important, but does not add much insight. Indeed, only a single paragraph is devoted to this case in the Results section. Given the similarity between Figs. 6 and 7, I ask the authors to consider moving Fig. 7 from the main text to the Supporting Information section.

Minot comments:

1. None of the authors seem to be associated with affiliation 3.

2. Five lines before the end of the Abstract: typo, "and did".

3. Line 24: was � has been

4. Line 36: "a a"

5. Line 68-69: "The essence of multiplexing…" This is a very important sentence that should appear earlier, in the Abstract and Introduction sections.

Reviewer #2: Review is uploaded as an attachment. Just in case a text version here:

The authors discuss how the synaptic weights from two the presynaptic populations with different frequencies are changed under the STDP. They employ an intricate Fokker-Plank formalism to solve the evolution of weights under the assumption of the time-scale separation between neuronal and synaptic dynamics. Authors derive boundaries for the stability of the rhythmic and winner-take-all eigenvalues to set values of parameters for the numerical simulations of the conductance-based neuronal model for numerical simulations.

\\subsection*{Major}

The main problem with the paper is that the results are not yet strongly interpretable in neuroscience terms. The authors claim that they show how STDP \\emph{``enables the transmission of rhythmic activity downstream''}; however, they demonstrate only that the weights are not converging to any stable fixed point. The continuous change of the synaptic strength seems to signify that the mini-network is possibly not learning anything.

If authors would show some cases where this continuous weight dynamics would indeed serve to learn something, it will make the neuroscience case much stronger.

Maybe if the phases of the neurons for different frequencies would be differently distributed (more or less uniformly, possibly randomly, but correlated)? How strong the results depend on the uniform distribution of the phases?

There could be, for example, some information transfer between the population phase and firing of the output neuron. Or maybe under a structured input there could be a non-trivial stable weights configuration.

It would be very useful if the authors could explain how the dynamics they observe can be interpreted as multiplexing of information (also adding on how they would define it in this case).

Another major problem with the paper is that the computations are notoriously hard to follow. Considering that they constitute the substantial part of the paper, I would suggest explaining some transitions more in the supplementary, adding the intuitive explanation, when necessary repeating the previous papers. I believe that after studying in detail the earlier publications of the senior author (Luz and Shamir, 2016, 2014, 2012), I could get a better understanding of the mathematics involved. However, the paper should be reasonably self-contained to be generally comprehensible without a long investigation of the other manuscripts.

\\subsection*{ Intermediate}

\\item Why the Eq. 1 represents a mean rate? Average over what? I imagine that it is an equation describing the inhomogeneous rate.

\\item I think the brackets in Eq. 2b are misplaced ($\\delta_{\\eta, \\xi}$ should be outside of the brackets)

\\item The $\\delta(\\Delta t)$ is not defined in Eq.~4. It cannot be the Dirac-$\\delta$ (as there should be a value for every $\\Delta t$), so I am slightly puzzled.

\\item the figure with the STDP kernels would be helpful

\\item The Fig. 1 is very beautiful but can come much earlier in the text. It also has the same basic frequency for red and blue populations, though the claim is that they are different. And overall, the red neurons spike-trains are identical copies of the blue ones. It is easy to fix, and it would then be more consistent.

\\item For the understanding of the results concerning the stability of rhythmic mode would be interesting to add the figures for the synaptic dynamics (similar to 5 and 6) in a qualitatively different parameter regime.

\\subsection*{Minor}

\\begin{enumerate}

\\item Fig 2,3,4 are very hard to read, as the lines are very fine, and the numbers are tiny.

\\item In the caption of (c) it is written $\\mu \\in (0,1)$ but you probably wanted to write $\\mu \\in [0,1)$.

\\item Authors name and surname are in the wrong order

\\end{enumerate}</t2<<3.>

**Have all data underlying the figures and results presented in the manuscript been provided?**

Reviewer #1: Yes

Reviewer #2: Yes

PLOS authors have the option to publish the peer review history of their article (what does this mean?). If published, this will include your full peer review and any attached files.

Reviewer #1: No

Reviewer #2: No
---

## [Decision Letter · Decision Letter 1]

20 May 2020

Dear Mr. Sherf,

Thank you very much for submitting your manuscript "Multiplexing rhythmic information by spike timing dependent plasticity" for consideration at PLOS Computational Biology. As with all papers reviewed by the journal, your manuscript was reviewed by members of the editorial board and by several independent reviewers. The reviewers appreciated the attention to an important topic. Based on the reviews, we are likely to accept this manuscript for publication, providing that you modify the manuscript according to the review recommendations.

Sincerely,

Jonathan Rubin

Associate Editor

PLOS Computational Biology

Lyle Graham

Deputy Editor

PLOS Computational Biology

[LINK]

Reviewer's Responses to Questions

**Comments to the Authors:**

Reviewer #1: The authors have successfully addressed all my comments.

Reviewer #2: I can congratulate the authors; the manuscript has improved in readability significantly. However, on this path, there are still some things to do that can make reading more enjoyable.

I agree with reviewer 1, the scientific merits of the presented work will allow it to be published in PLOS CB. Nevertheless, I suggest investing time in editing still and making the scientific results accessible to the reader.

I appreciate adding the Fig.2, panel a) probably should have Delta K on the x-axis. The figure also could be larger; there is no need for panels to be smaller than legends. For both panels, not indicated in the legend tau could be listed in the caption (tau_ for (a), tau_+ for (b))

Some things might be formulated more precisely, allowing the reader to understand what is happening. For example, what does “limited” mean in the context of: “Our theory reveals that the winner-take-all like competitions between the two populations is limited. ”

It will also help to explicitly state what “homogeneous solution” means (153) (that all the weight from input neurons of different preferred phases and different populations are the same).

Figure 3. helped a lot with the understanding of the different options; it is central for the presentation of the result and could be made larger. It could also become more informative by indicating that frequencies with non-zero powers observed in the panels c), f), and i) correspond to the frequencies of the two populations (maybe indicate particular frequencies with slim lines?).

What are the third non-zero frequency arising in the panels (f) and (i)? Does its presence in panel i) allows reconstructing the “lost” input frequency. In the caption: “Synapses (traces) are differentiated by color according to their preferred phase.” The preferred phase belongs not to the synapse but the presynaptic neuron.

Phase diagrams have a very small description font. Possibly the descriptions can be united, and the font enlarged.

I did not find an explanation of what is the function delta(x) (without indices as seen in Eq. 4 and 5). It seems to be an indicator function of 0.

Instead of Spikes/sec - Hz. On one occasion, misprinted “mses” instead of sec. Generally, as a unit, “s” typically stands for seconds.

**Have all data underlying the figures and results presented in the manuscript been provided?**

Reviewer #1: Yes

Reviewer #2: Yes

PLOS authors have the option to publish the peer review history of their article (what does this mean?). If published, this will include your full peer review and any attached files.

Reviewer #1: No

Reviewer #2: No
---

## [Editor Report · Decision Letter 2]

29 May 2020

Dear Mr. Sherf,

We are pleased to inform you that your manuscript 'Multiplexing rhythmic information by spike timing dependent plasticity' has been provisionally accepted for publication in PLOS Computational Biology.

Best regards,

Jonathan Rubin

Associate Editor

PLOS Computational Biology

Lyle Graham

Deputy Editor

PLOS Computational Biology

---

## [Editor Report · Acceptance letter]

19 Jun 2020

PCOMPBIOL-D-19-02160R2 

Multiplexing rhythmic information by spike timing dependent plasticity

Dear Dr Sherf,

I am pleased to inform you that your manuscript has been formally accepted for publication in PLOS Computational Biology. Your manuscript is now with our production department and you will be notified of the publication date in due course.

With kind regards,

Sarah Hammond
